# Benefits and harms of copyright restrictions and conditions on burnout and other psychometric assessment scales

Robert G. Badgett [1]*, Sudha Xirasagar[2], Hayrettin Okut[3]

**1** Department of Internal Medicine, University of Kansas School of Medicine-Wichita, Wichita, Kansas, United States of America, **2** Department of Health Services Policy and Management. Arnold School of Public Health, University of South Carolina, Columbia, South Carolina, United States of America, **3** Department of Population Health and Office of Research, University of Kansas School of Medicine-Wichita, Wichita, Kansas, United States of America

* rbadgett@kumc.edu

## Abstract

### Introduction

Copyright conditions can hinder the scientific and usage impact of essential assessment tools in healthcare, organizational psychology, and other fields. The stories of the Maslach Burnout Inventory (MBI) and the Mini-Mental Status Examination (MMSE) are well-documented. Despite decades-old recommendations to blend copyright restrictions of intellectual property when used for commercial purposes with permissive but distinct copyright terms for non-commercial and scholarly use, few authors of assessment tools do so.

### Objective

To assess whether restrictive copyrights limit the long-term scientific impact of copyright holders (often medical schools/faculty), by inhibiting downstream use, research, and improvement of measurement tools by other researchers.

### Methods

We compared MEDLINE citation trends of two restrictive surveys (requiring royalty payments, prohibiting item modifications) – the Maslach Burnout Inventory (MBI) and the Mini-Mental Status Examination (MMSE) – against four free-to-use and adapt surveys. Our outcome was the annual, theme-relevant citation ratio of each survey since publication. The ratio was the number of publications featuring the scale name in the title or abstract divided by total publications addressing the original theme (purpose) of the scale. We performed paired comparisons of the most recent slopes of the annual citation ratio, pairing each restrictive scale with a permissive scale, totaling eight comparisons.

**Data availability statement:** https://osf.io/gfkxc/ https://doi.org/10.17605/OSF.IO/GFKXC.

**Funding:** The author(s) received no specific funding for this work.

**Competing interests:** The authors have declared that no competing interests exist.

## Results

The restrictive MBI and MMSE initially showed rising slopes of annual citation ratios, and a decline in later years with negative slopes. Further, the final slope segments for all four permissive surveys were positive, vs. negative for the restrictive surveys, the slope difference being statistically significant for six out of eight comparisons.

## Conclusions

Restrictive copyright conditions on assessment scales risk undermining their long-term scientific and functional relevance. Previously recommended blended copyright approaches can facilitate survey creators' revenues via commercial use royalties, while sustaining continued scholarly development and noncommercial usage to improve long-term relevance.

## Introduction

Surveys and other assessment scales often have copyright restrictions, requiring royalty payments as a condition of use, regardless of commercial or scholarly purposes, and/or prohibiting adaptations and derivative works (modifications). These restrictions have adversely affected medical care and human resource management initiatives for employee well-being. A current example is the ample use of the open-access with permissive reuse single-item burnout question (SIBOQ) that is less accurate than validated scales that are copyright-restricted.[1] Another example was practitioners' difficulty in accessing a validated scale to assess patient cognition after the Mini-Mental Status Examination (MMSE) scale introduced royalties and prohibition of survey item modifications.[2,3]

For owners who prefer to restrict access to their intellectual property, in 2011, a commentary in the New England Journal of Medicine by Newman and Feldman promoted the then-emerging blended copyright approach.[3] Rather than an either-or approach (completely permissive or restrictive), the blended option supports continued use for noncommercial purposes and scholarly advancement based on automatic, permissive licensing for these purposes, while permitting scale creators to monetize their intellectual property when used commercially through separate licensing at request for commercial users. In reality, however, this approach to alternative licensing for survey research appears to have remained dormant. For example, the resources hosted by the Open Science Framework website (https://osf.io/search) offer a dropdown list of licensing types along with the number of licenses against each type. Currently, less than 10% of instruments/products feature a noncommercial (NC) use or share-alike (SA) option, both indicators of a blended approach.

The paucity of blended licensed surveys/products on the OSF and similar websites suggest that content creators are assuming that copyrighting is a binary decision – either full access or restricted access. This perception may be reinforced by most journals requiring authors to either transfer copyright ownership to the journal, or pay

for unrestricted, open access publication. While studies have examined motivation of software developers [4], there are no studies on the motivations driving copyright decisions for assessment tools. The latter could be on shaky ground because it is likely that these decisions are driven by content creator's employers rather than by the creators.

Restrictive copyrights may harm scale owners. The Maslach Burnout Inventory (MBI) experience presents a case in point. Widespread use of the SIBOQ by researchers (including by authors of the current study), [5,6] omission of questions on burnout in many employee surveys (including those fielded by major employers), [7–9] and the consequent challenges of comparing burnout-related findings across surveys, [1] suggest that the original MBI scale may be missing both revenue opportunities and potential frontrunner status due to its restrictive copyright terms. Ongoing research into alternative measures of burnout [10–12] may portend further decline in its use, and eventual supersession by other surveys. In all fairness, the MBI and MMSE were developed in an era that prioritized monetizing of intellectual property: intellectual property rights protection in the U.S. underwent multiple extensions of copyright durations, [13] and the Bayh-Dole Act extended patent ownership privileges to universities.[14] Open-access scholarly content via platforms such as PubMed Central and Creative Commons did not exist. It was a time when copyright options were seen as a binary choice – either restricted or public domain.[15]

An additional consequence of the decline of well-validated scales is loss of construct validity (e.g., "burnout") due to abandoning the standardized measure of the original construct either due to loss of comparability across different participant samples studied with competing scales, or across longitudinal measures in a sample, over time, if different scales were used in later rounds to circumvent the cost implications of copyright restrictions. Widely accessible, standardized measures enable organizations and clinicians/researchers to track trends over time and benchmark their results against larger or other populations. Lacking such measures with widely accessible scales, surveys of workforce well-being either omit burnout questions or use alternative survey items, resulting in client organizations interpreting measurements done with unvalidated scales as they deem best.

Our goal was to compare the duration and intensity of research activity that used surveys/scales with restrictive copyright terms versus research activity that used surveys with permissive copyright terms. We compared the trends of the theme-relevant, yearly citation ratios of fully copyright-restricted surveys/scales with surveys having a blended approach to their copyright terms. We hypothesized that a highly restrictive copyright directly hurts the copyright owners' interests by inhibiting the continuation of research by other scholars that could extend the scientific impact and longevity of the original survey. We hypothesized that our findings would illustrate the real-world impact of a blended copyright approach for copyright owners that would permit continuation of intellectual property revenues while enabling further improvement and continued relevance and use of their scale.

## Methods

We studied the MEDLINE citations of six surveys from their inception through 2023.

### Surveys and scales studied

Two surveys, the MMSE and Maslach Burnout Inventory (MBI), require royalty payments or prohibit item modification as conditions of use. Two comparison surveys without these restrictions were identified: the Saint Louis University Mental Status (SLUMS) and Patient Health Questionnaire (PHQ-9). Informed by the comparative findings on these four surveys, we chose to explore the long-term research use of two open-access clinical care scales, the neonatal health status scale, APGAR, and the mental health scale used to screen for alcoholism, CAGE (Cut down, Annoyed, Guilty, Eye-opener).

The search terms used are in the Table in S1 Table. We limited our searches to the Medline subset of PubMed in order to ensure a more stable set of journals for comparisons across different timelines of the study surveys, and to avoid bias against using MeSH terms when needed for the search. To qualify as a citation of a scale, the scale title had to be featured in the paper title or abstract, which, in the aggregate, should validate that the research activity was indeed anchored

in the survey and focused on the original purpose of the survey. The annual citation ratio for each survey (defined below) allows comparisons across surveys and over time, including expansion to new usage settings or purposes, the latter likely to be reflected in an abrupt increase in the citation ratio.

### Outcome measure

The outcome studied was the theme-relevant citation ratio each year following survey publication through 2023. The ratio was defined as the number of citations featuring the survey or scale name in the title or abstract divided by total publications featuring the core purpose of the survey when it was first developed. For all surveys, the ratio was multiplied by 100 to facilitate graphic display. For example, the ratio for the MMSE was the number of citations per year for 'Mini-Mental status'[TIAB] divided by the number of citations per year for dementia [TIAB]. The search terms used to develop each scale's citation ratio are those listed in the Table in S1 Table.

The ratio estimates the proportion of all research studies on the constructs initially targeted by the survey that actually used the original survey. Notably, the citation ratio is expected to rise abruptly when the use of the survey expands to new research contexts and questions. Conversely, the ratio should show a significant fall when a survey is little cited in relation to its initial purpose and study population setting. An illustration of the calculation and graphic display of the ratio over time is presented in Figure in S1 Fig.

### Statistical methods

Our principal measure was the most recent or final segment slope in the spline regression curves of the annual survey citation ratio plot of each survey since its first publication up to (and including) 2023.

We used natural cubic spline regression (adjusted for covariates) to generate non-linear models to accommodate discontinuous changes in the slope of the citation ratio over time. We chose the optimal number of interior knots, or deflections, in the spline plots by using ANOVA to compare regressions with 0–9 interior knots using the ANOVA tests and adjusted $R^2$ values. We selected as optimal the number of knots beyond which no further improvement of model fit was achieved ($p > 0.05$ for the ANOVA test and no improvement in the adjusted $R^2$). An illustration of the selection procedure with the Maslach Burnout Inventory to determine the optimum number of knots and identify the optimal model is shown in the Figure in S2 Fig

and in an online video using the same data. [16]

For our primary outcome, we compared survey slopes over the years represented in the final segment of each spline regression analysis. Additional methodological details are presented in the Figure in S1 Fig and in the Figure in S2 Fig. Our data and statistical analysis code are available at https://osf.io/gfkxc/.

### Results

The slopes of the final spline models in the covariate-adjusted plots of the annual citation ratios for MMSE, MBI, SLUMS, and PHQ-9 are presented in Table 1. For the two copyright-restricted surveys (MBI and MMSE), rising citation ratios in the early years are observed, followed by declining ratios with time (Fig 1, Panels A and B). The slopes of the final spline models of the two permissive-use surveys, the SLUMS and PHQ-9 (Fig 2, Panels A and B) show a continuing rise, with the slopes of the final time-trend segments being positive. The two permissive-use surveys' citation ratio slopes were significantly greater than the corresponding slopes of the two copyright-restricted surveys in three of the four paired comparisons (Table).

The long-established, permissive APGAR and CAGE scales show a continuing rise in citation ratios, although CAGE had an interim period of negative slope (Figure in S3 Fig). The final slopes of the CAGE and APGAR scales were also significantly greater than those of the two copyright-restricted surveys in three of the four comparisons (Table).

**Table 1. Summary of findings. In six of the eight comparisons, the slope of the permissive survey was significantly higher than the slope of the restrictive survey.**

| Survey or scale | First published | Copyright enforcement started | Final segment slope (95% CI) | Significance testing of differences in slopes |
|---|---|---|---|---|
| **Restrictive usage** | | | | |
| MMSE* | 1975 | 2001 | −0.047 (−0.138 to 0.045) | Compared to: PHQ-9: 0.000 SLUMS: 0.278 APGAR: 0.000 CAGE: 0.272 |
| MBI† | 1981 | 1996 | −0.675 (−1.22 to −0.123) | Compared to: PHQ-9: 0.000 SLUMS: 0.016 APGAR: 0.003 CAGE: 0.015 |
| **Permissive usage** | | | | |
| PHQ-9‡ | 1999 | NA | 2.409 (2.106 to 2.711 | See above |
| SLUMS§ | 2006 | NA | 0.004 (0.002 to 0.006) | See above |
| **Permissive copyrights with long-term use** | | | | |
| APGAR§ | 1953 | NA | 0.173 (0.132, 0.214) | See above |
| CAGE¶ | 1974 | NA | 0.005 (−0.007 - 0.017) | See above |

Notes:

\* Mini-Mental State Examination

† Maslach Burnout Inventory

‡ Patient Health Questionnaire – 9. This is used to aid the diagnosis of depression.

§ Saint Louis University Mental Status

|| Appearance, pulse, grimace, respiration. This is used to assess the health of newborns.

¶ Cut down, annoyed, guilty, eye opener. This aids the diagnosis of alcoholism.

Notably, the plot of the CAGE Questionnaire shows a non-linear pattern, with a second rise in the citation ratio after 2020 related to an allied topic but distinct from the original topic of the survey that fueled the pre-2020 citation ratios. The search terms in Table in S1 Table show that all the initial citations stemmed from the "Alcohol Problem" section among the Medical Subject Headings (MeSH), the original topic of the survey published by Mayfield et al.[17] Our review of topics involved in the second rise in the post-2020 citation ratio identified the new influence of the 'Drinking Behavior' MeSH term within the Behavior section of the MeSH tree in the articles captured by the search. We verified that this was indeed the case by a second PubMed search requesting exclusion of "Alcohol Problem":

(CAGE Question*[TIAB] NOT 'Alcohol Problem'[MeSH]) AND MEDLINE[SB]

Just as the CAGE scale was newly used to study at-risk drinking behaviors among persons *without* diagnosed alcoholism, we found the PHQ-9 was recently used to capture evolving depressive symptoms in patients *without* a diagnosis of depression.

(PHQ-9[TIAB] NOT 'Depressive disorder'[MeSH]) AND MEDLINE[SB]

## Discussion

In six out of eight comparisons of permissive-use versus copyright-restricted surveys or scales, the former had significantly higher citation ratios that sustained over time since first publication compared to the copyright-restricted surveys.

## Panel A. The Maslach Burnout Inventory.

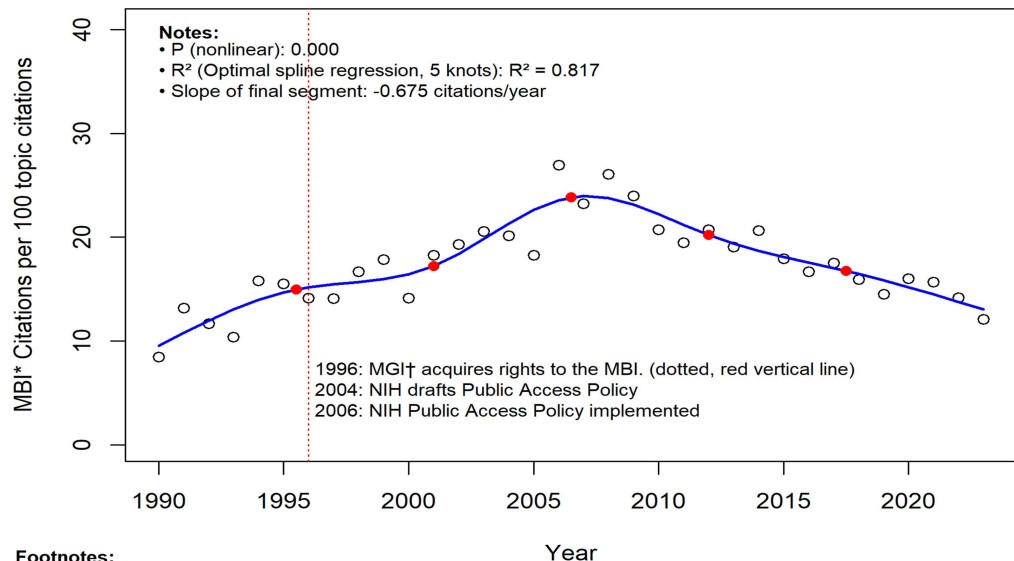

**Notes:**
- P (nonlinear): 0.000
- R² (Optimal spline regression, 5 knots): R² = 0.817
- Slope of final segment: -0.675 citations/year

1996: MGI† acquires rights to the MBI. (dotted, red vertical line)
2004: NIH drafts Public Access Policy
2006: NIH Public Access Policy implemented

**Footnotes:**

* MBI. Maslach Burnout Inventory. First published: Maslach, 1981. DOI: 10.1002/job.4030020205

† MGI Mind Garden, Inc.

## Panel B. The Mini-mental Status Exam.

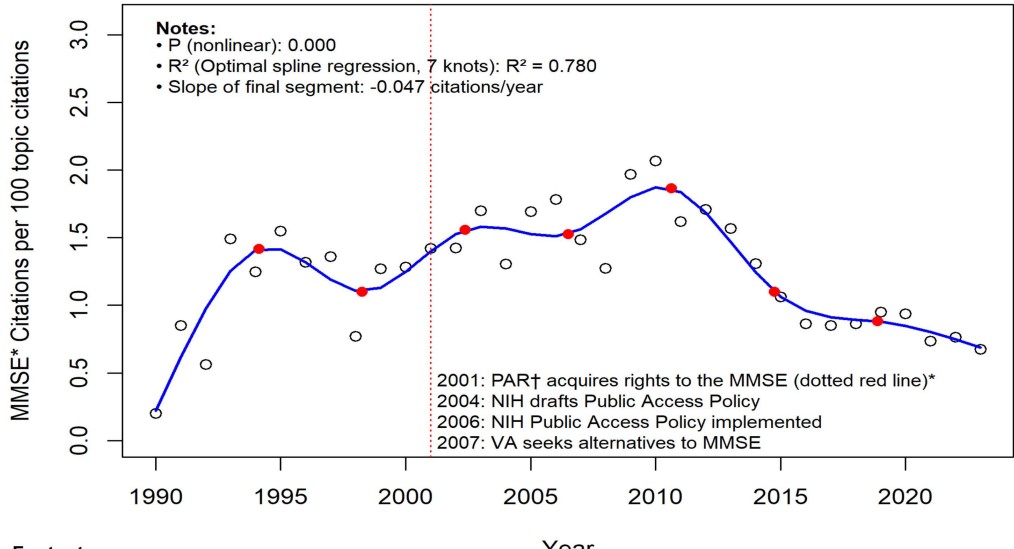

**Notes:**
- P (nonlinear): 0.000
- R² (Optimal spline regression, 7 knots): R² = 0.780
- Slope of final segment: -0.047 citations/year

2001: PAR† acquires rights to the MMSE (dotted red line)*
2004: NIH drafts Public Access Policy
2006: NIH Public Access Policy implemented
2007: VA seeks alternatives to MMSE

**Footnotes:**

* MMSE. Mini-mental Status Exam. First published: Folstein, 1975. PMID: 1202204

† PAR, Inc. Psychological Assessment Resources.

**Fig 1. Two surveys with restrictive copyrights.**

## Panel A. The Patient Health Questionnaire.

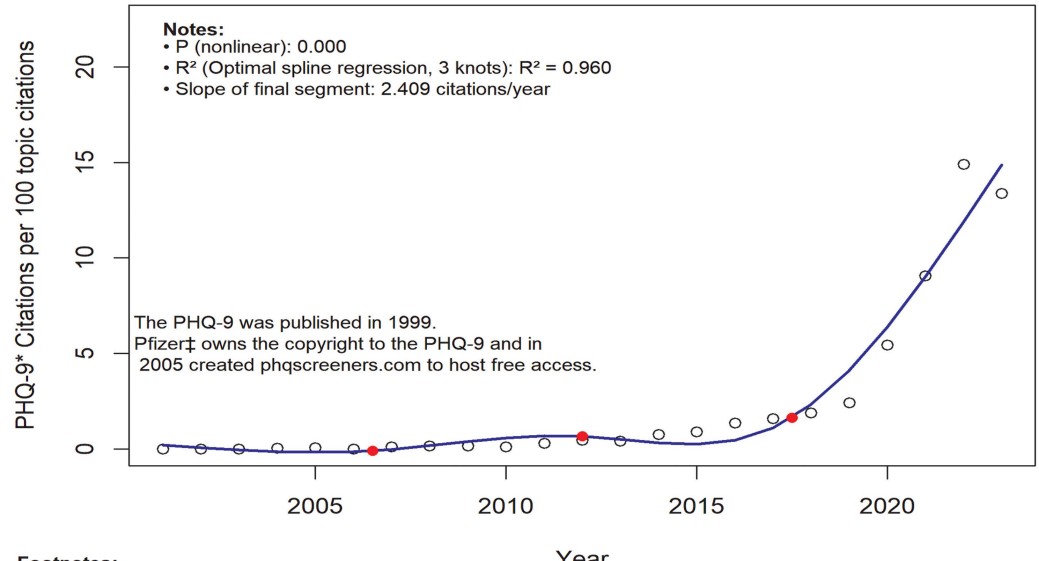

**Notes:**
• P (nonlinear): 0.000
• R² (Optimal spline regression, 3 knots): R² = 0.960
• Slope of final segment: 2.409 citations/year

The PHQ-9 was published in 1999.
Pfizer‡ owns the copyright to the PHQ-9 and in
2005 created phqscreeners.com to host free access.

**Footnotes:**
* PHQ-9. Patient Health Questionnaire-9. First published: Spitzer, 1999. PMID: 10568646
† Pfizer Inc.

## Panel B. The Saint Louis University Mental Status Exam.

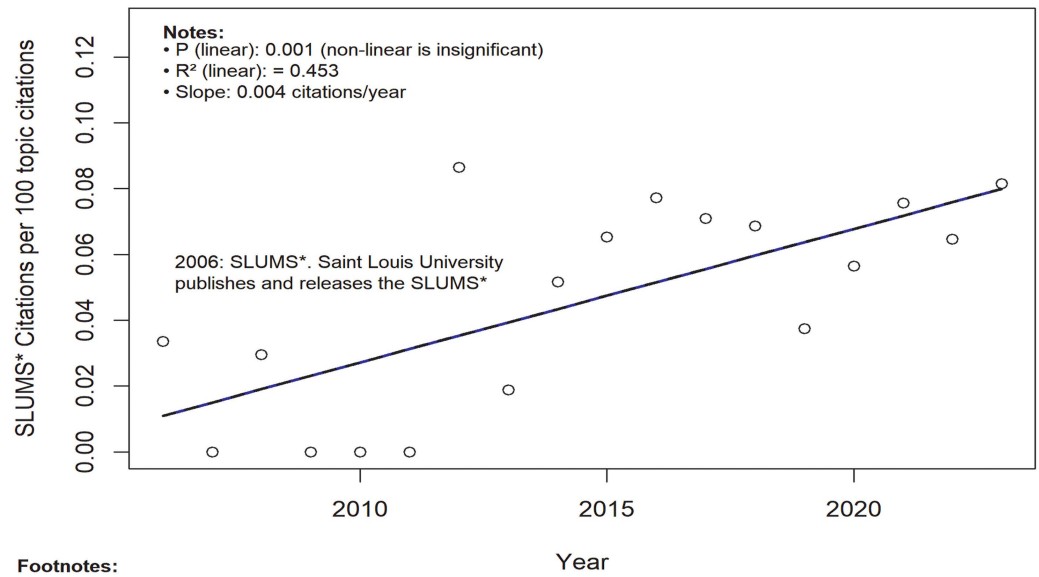

**Notes:**
• P (linear): 0.001 (non-linear is insignificant)
• R² (linear): = 0.453
• Slope: 0.004 citations/year

2006: SLUMS*. Saint Louis University
publishes and releases the SLUMS*

**Footnotes:**
* SLUMS. Saint Louis University Mental Status Exam. First published: Tariq, 2006. PMID: 17068312

**Fig 2. Two surveys with permissive copyrights.**

This result is consistent with the findings on a similar topic of open-access scholarly publications – these publications had higher rates of downloads and citations than copyright-restricted journal articles.[18–20] We are not aware of prior studies that specifically examined published evidence of the use rates of healthcare assessment tools by user access to the survey. Not finding precedents of assessment tool usage frequency stratified by copyright type, we used journal citations as a measure of ongoing tool usage.

Our finding on the adaptation of permissive surveys to measure phenomena in new populations, or to study new research questions needs more work to explore why the re-purposing of surveys to new contexts and circumstances occurs. For example, some surveys showed decreasing usage about the time of the Great Recession of 2009. The COVID-19 pandemic may also have affected the contexts or settings available to screen for depression and alcohol drinking behavior.

Our study has a limitation that may affect some Medline searches with limited recall capacity causing loss of capture of potentially relevant papers. Because our search was limited to the scale name appearing in the article title and abstract, we may have missed studies that mentioned the survey title only in the article text. However, reasonably speaking, if the survey was a core component of the study, it would be unlikely to miss being mentioned in the abstract. While older or widely used surveys may be at risk of not being mentioned by name in the abstract, this limitation is mitigated by our finding on older, permissive surveys showing continued upward trends in their citation ratios based on appearance of the scale name in the abstract (S3 Fig). On the positive side, our search precision is likely to be high given that all the surveys studied have unique names. The exception would be studies that used the word "cage", for example, referring to the "rib cage" instead of the "CAGE" survey. However, this scenario is very unlikely as we searched for "cage question*" in the title and abstract and also required the occurrence of our second search term, "alcohol problem [MeSH]" to capture the study. In addition, all search strategies are tethered to a fixed year of first use because all our candidate surveys were published when they were first developed. This allowed our search strategies to map accurately to the MeSH terms initially assigned for surveys' topics by the National Library of Medicine.

Another limitation is distinct differences in scale user profession and setting of the instruments compared: APGAR and CAGE instruments are used in different types of clinical settings and participant types, which in turn differ from management science/organization behavior scales typically used by researchers. These differences may affect their citation rate due to extraneous reasons other than copyright conditions, such as the frequency of use in original research studies. While this remains a structural limitation of the comparisons presented, the consistency of findings in all paired comparisons of permissive with restricted instruments supports a role of copyright limitations in scale use and evolution.

Our findings suggesting limited durability of copyright-restrictive surveys in the research landscape, in contrast to permissive surveys which additionally, gain new purposes for use with time [21], have practical implications for content creators. It dispels the perception of an either-or compulsion, and provides evidence for the benefits of a middle-of-the-ground, blended approach - dual licensing which preserves commercial revenue and concurrently facilitates non-commercial user access as described by Newman and Feldman.[3] Authors can foster ongoing survey or scale development and thus earn an enduring scientific impact of their original work while optionally monetizing their intellectual property through copyright charges on commercial users. Authors may consider using a permissive Creative Commons (license) CC BY-NC-SA or similar mechanism that permits survey derivatives and modifications in noncommercial settings. Content owners could, in parallel to displaying copyright relaxations for noncommercial use, display the terms and pricing for commercial users including derivative permissions and research licensing, or simply display contact information for the latter types of users.[22] Survey authors may negotiate with potential commercial license administrators that permission to use for research purposes will be royalty-free and non-exclusive. Free use for research purposes is, incidentally, aligned with the National Institutes of Health (NIH) Data Sharing Policy that applies to products of NIH grant-funded research.[23]

For content creators considering a blended approach to copyright protection, the proposed process above is not unfettered, due to potential restrictions imposed by the copyright policies of most journals. As noted earlier, most journals do

not readily support a blended approach. To implement a blended approach to copyrighting in either of these settings, the authors could consider releasing their survey as a "project" on the Open Science Framework (OSF) website prior to submission to the journal. The OSF supports authors to publish a survey with a variety of copyright options. Our assessment is that survey/scale authors may benefit from exploring copyright options with their academic medical center's intellectual property office or research administration. One caveat is that while the creator of intellectual property may desire their creation to be open-access [24], that may not be the preference of their employer who may control the copyright decision. Furthermore, it is very possible that legal counsel of employers may be unaware of the nuances of blended copyrights in assessment scale publishing.

An alternative approach for survey authors is to publish as a preprint with permissive copyright prior to submission to a journal.[25] Not all journals will accept articles with preprints.[26] While easy to deposit a preprint, subsequent modifications to the survey may not be legally bound by the copyright terms intended by the original authors. Regardless, using the OSF, while more complicated, facilitates updates to copyrighted property (S1 Text).

Studies show that it takes 10–20 years for an innovation in healthcare to manifest in clinical practice.[27] PubMed Central and the Creative Commons were launched in 2000 and 2001, respectively, to accelerate science and improve practice. At the quarter century mark, the current timing appears appropriate for researchers to harness the changing framework of copyright practices and empower their innovations to exert the maximum impact on science and health.

## Conclusion

Restrictive survey copyright conditions on healthcare assessment scales may undermine their long-term scientific and clinical relevance. We encourage scholars who are developing scales to consider newer blended copyright approaches as recommended by Newman and Feldman.[2,3]

## Supporting information

**S1 Table. Search terms used for MEDLINE for each scale.**
(DOCX)

**S1 Figure. Example using the Maslach Burnout Inventory to show the calculation of the citation ratio and its resulting plot.**
(DOCX)

**S2 Figure. Example using the Maslach Burnout Inventory to show the selection process to identify the optimal number of knots (deflections) in the regression plots.**
(DOCX)

**S3 Figure. Two surveys/scales with permissive copyrights and long-term follow-up possible.**
(DOCX)

**S1 Text. Publishing a survey with a blended license at the Open Science Framework.**
(DOCX)

## Author contributions

**Conceptualization:** Robert G. Badgett, Sudha Xirasagar, Hayrettin Okut.

**Data curation:** Robert G. Badgett, Hayrettin Okut.

**Formal analysis:** Robert G. Badgett, Hayrettin Okut.

**Investigation:** Robert G. Badgett.

**Methodology:** Robert G. Badgett.

**Project administration:** Robert G. Badgett.

**Software:** Robert G. Badgett.

**Supervision:** Sudha Xirasagar.

**Validation:** Robert G. Badgett.

**Visualization:** Robert G. Badgett.

**Writing – original draft:** Robert G. Badgett.

**Writing – review & editing:** Robert G. Badgett, Sudha Xirasagar, Hayrettin Okut.

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
