## [Decision Letter · Decision Letter 0]

4 Mar 2026

PONE-D-25-65524Benefits and harms of copyright restrictions and conditions on burnout and other clinical assessment scales

PLOS One

Dear Dr. Badgett,

Thank you for submitting your manuscript to PLOS ONE. After careful consideration, we feel that it has merit but does not fully meet PLOS ONE’s publication criteria as it currently stands. Therefore, we invite you to submit a revised version of the manuscript that addresses the points raised during the review process.

What seems to be important to both reviewers is making the context of your research clearer to the audience from other research fields. The readership pool is potentially much wider than you assumed, and some explanations could improve the text. Another important issue raised by the reviewers is to what extent the licensing model is a decisive factor.

We look forward to receiving your revised manuscript.

Kind regards,

Piotr Stec

Guest Editor

PLOS One

Journal Requirements:

1.Please ensure that your manuscript meets PLOS ONE's style requirements, including those for file naming. The PLOS ONE style templates can be found at https://journals.plos.org/plosone/s/file?id=wjVg/PLOSOne_formatting_sample_main_body.pdf and

Reviewers' comments:

Reviewer's Responses to Questions

**Comments to the Author**

1. Is the manuscript technically sound, and do the data support the conclusions?

Reviewer #1: Yes

Reviewer #2: Yes

2. Has the statistical analysis been performed appropriately and rigorously?

Reviewer #1: Yes

Reviewer #2: Yes

3. Have the authors made all data underlying the findings in their manuscript fully available?

Reviewer #1: Yes

Reviewer #2: Yes

4. Is the manuscript presented in an intelligible fashion and written in standard English?

Reviewer #1: Yes

Reviewer #2: Yes

5. Review Comments to the Author

Reviewer #1: I have reviewed the manuscript entitled Benefits and harms of copyright restrictions and conditions on burnout and other clinical assessment scales. This manuscript evaluates whether copyright restrictions applied to widely used clinical assessment tools may reduce their long term scientific impact by limiting reuse adaptation and scholarly development. The authors compare MEDLINE citation trends for two copyright restricted scales against four freely usable instruments using a theme relevant annual citation ratio and spline regression models through 2023. The central finding is that copyright restricted scales demonstrate declining citation ratios over time compared with permissively licensed instruments which show sustained or increasing use. The authors argue that restrictive licensing may undermine long term relevance and propose a blended copyright model permitting non commercial scholarly reuse while maintaining commercial licensing rights. The topic is timely and important particularly in light of the increasing emphasis on open science and research tool accessibility.

The manuscript addresses an important and underexplored policy issue namely the downstream scientific impact of licensing restrictions placed on measurement tools used in healthcare research. The proposed conceptual mechanism in which restrictive licensing discourages derivative scholarship and thereby reduces long term uptake is logical and policy relevant. In addition the theme relevant annual citation ratio represents an innovative attempt to normalize research adoption across heterogeneous time horizons and the spline regression modeling approach described in the Methods section appears appropriate for evaluating non linear adoption trajectories across decades.

However there are several important methodological and interpretive concerns that should be addressed. The primary outcome assumes that the presence of a scale name in the title or abstract reflects meaningful research use as described in the Methods section. In practice many studies employ validated tools without naming them in the abstract and derivative instruments or modified versions may not explicitly reference the original scale name. Furthermore citation practices vary substantially across journals and fields. As a result the selected metric may introduce systematic bias against older or widely internalized instruments whose usage becomes implicit over time. Although the authors acknowledge limited recall associated with title and abstract searches in the Discussion section this limitation may be more central to the validity of the study’s inference than is currently suggested.

Relatedly the attribution of causality may be overstated in its present form. Observed declines in citation ratios for restricted tools may reflect construct drift evolving definitions of burnout or cognition the emergence of competing theoretical frameworks methodological shifts toward ultra brief screening tools or broader disciplinary trends such as post pandemic measurement priorities rather than copyright conditions per se. The manuscript would benefit from a more cautious causal framing or from sensitivity analyses that address innovation cycles construct fragmentation or time to replacement dynamics.

The selection of comparator instruments also warrants further justification. The inclusion of APGAR and CAGE introduces heterogeneity in construct type clinical context temporal diffusion patterns and baseline institutional adoption that may limit interpretability of licensing policy effects. These instruments differ substantially from burnout or cognitive screening tools and may therefore not represent ideal comparators in this analysis.

In addition there is a terminology issue that should be addressed for scholarly tone and precision. The Abstract and Introduction use the phrasing “restrictive copyrighting … with permissive copyrighting (copylefting)”. While grammatically intelligible the term “copylefting” is an informal neologism rather than a standard legal or scholarly term. In academic publishing this wording may appear advocacy oriented or imprecise. The authors may wish to consider replacing this phrasing with terminology such as “permissive licensing” “copyleft licensing” or “Creative Commons based licensing for non commercial scholarly use” in order to improve clarity and stylistic neutrality without altering meaning.

There are also several minor issues that should be corrected. The Mini Mental Status Examination appears with inconsistent capitalization in several locations and the phrase “Maslach Burnout Inventory(MBI)” is missing a space. Some policy discussion in the latter portion of the manuscript would benefit from citation to empirical implementation studies rather than commentary literature. In addition the estimate that fewer than ten percent of Open Science Framework projects included a noncommercial use statement would benefit from methodological clarification.

Overall this manuscript raises an important and policy relevant hypothesis but currently over attributes observed longitudinal citation trends to licensing conditions without sufficiently addressing alternative explanations. Clarification of terminology particularly replacing informal usage such as “copylefting” additional discussion of construct evolution and improved justification of comparator selection would strengthen both interpretability and scholarly tone. I therefore recommend major revision.

Reviewer #2: After reading the paper proposed I have a clear idea of the idea that authors pretend to expose and emphasize. The premise and the conclusions are clear, but sometimes is needed a little bit more of exposition and divulgation to explain the former importance of the copyright publications as a reason for their choosing. Furthermore, it's not clear the non-citation aspects of the election. Many times -and this is a serious problem in the present model of researching, dissemination and academic careers- the election of one review or another is focused not only or specifically in the impact but in the prestige -former or not- of each publication. Of course, citations are the way to consolidate the academic and scientific name of a review, but many times researchers are looking for a quick curriculum feedback instead of having a medium to long-term perspective.

6. PLOS authors have the option to publish the peer review history of their article (what does this mean?). If published, this will include your full peer review and any attached files.

**Do you want your identity to be public for this peer review?** For information about this choice, including consent withdrawal, please see our Privacy Policy.

Reviewer #1: **Yes:**Habil. Dr. Marlena Jankowska Augustyn, Assoc. Prof. (University of Silesia in Katowice), MBA (Deakin)

Reviewer #2: No

---

## [Author Response · Author response to Decision Letter 1]

16 Apr 2026

Editor’s comments:

1. making the context of your research clearer to the audience from other research fields

a. We made edits to the abstract, introduction, and title to encourage an audience beyond healthcare.

2. Another important issue raised by the reviewers is to what extent the licensing model is a decisive factor.

a. A new third paragraph in the introduction addresses this and notes there is little know about the content creator’s decision – and whether the creator or the creator’s employer makes the decision.

Reviewer 1:

1. I have reviewed the manuscript entitled Benefits and harms of copyright restrictions and conditions on burnout and other clinical assessment scales. This manuscript evaluates whether copyright restrictions applied to widely used clinical assessment tools may reduce their long term scientific impact by limiting reuse adaptation and scholarly development. The authors compare MEDLINE citation trends for two copyright restricted scales against four freely usable instruments using a theme relevant annual citation ratio and spline regression models through 2023. The central finding is that copyright restricted scales demonstrate declining citation ratios over time compared with permissively licensed instruments which show sustained or increasing use. The authors argue that restrictive licensing may undermine long term relevance and propose a blended copyright model permitting non commercial scholarly reuse while maintaining commercial licensing rights. The topic is timely and important particularly in light of the increasing emphasis on open science and research tool accessibility.

a. Thank you for your comment, no manuscript revision needed.

2. The manuscript addresses an important and underexplored policy issue namely the downstream scientific impact of licensing restrictions placed on measurement tools used in healthcare research. The proposed conceptual mechanism in which restrictive licensing discourages derivative scholarship and thereby reduces long term uptake is logical and policy relevant. In addition the theme relevant annual citation ratio represents an innovative attempt to normalize research adoption across heterogeneous time horizons and the spline regression modeling approach described in the Methods section appears appropriate for evaluating non linear adoption trajectories across decades.

a. Thank you for your comment, no manuscript revision needed.

3. However there are several important methodological and interpretive concerns that should be addressed. The primary outcome assumes that the presence of a scale name in the title or abstract reflects meaningful research use as described in the Methods section. In practice many studies employ validated tools without naming them in the abstract and derivative instruments or modified versions may not explicitly reference the original scale name. Furthermore citation practices vary substantially across journals and fields. As a result the selected metric may introduce systematic bias against older or widely internalized instruments whose usage becomes implicit over time. Although the authors acknowledge limited recall associated with title and abstract searches in the Discussion section this limitation may be more central to the validity of the study’s inference than is currently suggested.

a. We share this concern and on the surface our “metric may introduce systematic bias against older or widely internalized instruments whose usage becomes implicit over time” is a plausible concern. However, our findings that the older surveys (supplemental Figure S3) that the older surveys are continuing to have rising usage suggests that if this bias occurred, it was insufficient to obscure continued rising citation ratios in the older, permissive surveys. We added this observation to the limitation paragraph of our discussion.

4. Relatedly the attribution of causality may be overstated in its present form. Observed declines in citation ratios for restricted tools may reflect construct drift evolving definitions of burnout or cognition the emergence of competing theoretical frameworks methodological shifts toward ultra brief screening tools or broader disciplinary trends such as post pandemic measurement priorities rather than copyright conditions per se. The manuscript would benefit from a more cautious causal framing or from sensitivity analyses that address innovation cycles construct fragmentation or time to replacement dynamics.

a. We entirely agree with the reviewer. Construct drift with time, changing social mores/conditions, allied but distinct populations require evolving scales , and methodological shift towards tools that are shorter must be accommodated by “validated” scales but permitted to be modified within limits to address the need. These exigencies are very possible, and that is the point of our paper. We submit that, rather than being a limitation of our study, the examples offered by the Reviewer support our central theme that restrictive permissions harm survey evolution and prevent “drift” that exists in real life from being reflected in the measure, and prevent refinement of the survey to reflect distinctly different behaviors or allied populations or construct drift. An excellent example is how the UWES was refined by its creators to the UWES-3, which they kindly permit reusage. However, legally, that cannot happen with restrictive copyrights which reinforces our point .

b. To make the results and discussion more readable, We moved the second paragraph of the discussion to become the final paragraph of the results. This allowed grouping the two paragraphs that described investigations of secondary rises in the use of assessment tools.

5. The selection of comparator instruments also warrants further justification. The inclusion of APGAR and CAGE introduces heterogeneity in construct type clinical context temporal diffusion patterns and baseline institutional adoption that may limit interpretability of licensing policy effects. These instruments differ substantially from burnout or cognitive screening tools and may therefore not represent ideal comparators in this analysis.

a. We thank the reviewer for noting this potential limitation of the study. We have now added a limitation paragraph that reads as follows:

“Another limitation is distinct differences in scale user profession and setting of the instruments compared: APGAR and CAGE instruments are used in different types of clinical settings and participant types, which in turn differ from management science/organization behavior scales typically used by researchers. These differences may affect their citation rate due to extraneous reasons other than copyright conditions, such as the frequency of use in original research studies. While this remains a structural limitation of the comparisons presented, the consistency of findings in all paired comparisons of permissive with restricted instruments supports a role of copyright limitations in scale use and evolution”

6. In addition there is a terminology issue that should be addressed for scholarly tone and precision. The Abstract and Introduction use the phrasing “restrictive copyrighting … with permissive copyrighting (copylefting)”. While grammatically intelligible the term “copylefting” is an informal neologism rather than a standard legal or scholarly term. In academic publishing this wording may appear advocacy oriented or imprecise. The authors may wish to consider replacing this phrasing with terminology such as “permissive licensing” “copyleft licensing” or “Creative Commons based licensing for non commercial scholarly use” in order to improve clarity and stylistic neutrality without altering meaning.

a. We agree and deleted “copyleft” from the manuscript and modified the sections that had “copyleft”.

7. There are also several minor issues that should be corrected. The Mini Mental Status Examination appears with inconsistent capitalization in several locations and the phrase “Maslach Burnout Inventory(MBI)” is missing a space.

The spacing issue was corrected and the MMSE edits were made to only use “Mini-Mental”.

Some policy discussion in the latter portion of the manuscript would benefit from citation to empirical implementation studies rather than commentary literature.

Citations have been added, but we noted that the citations are to related issues and we do not find prior studies of our specific topic on assessment tools.

In addition the estimate that fewer than ten percent of Open Science Framework projects included a noncommercial use statement would benefit from methodological clarification.

a. Please see revisions to second paragraph of the introduction.

8. Overall this manuscript raises an important and policy relevant hypothesis but currently over attributes observed longitudinal citation trends to licensing conditions without sufficiently addressing alternative explanations. Clarification of terminology particularly replacing informal usage such as “copylefting” additional discussion of construct evolution and improved justification of comparator selection would strengthen both interpretability and scholarly tone. I therefore recommend major revision.

a. Thank you for summarizing.

Reviewer 2:

1. After reading the paper proposed I have a clear idea of the idea that authors pretend to expose and emphasize. The premise and the conclusions are clear, but sometimes is needed a little bit more of exposition and divulgation to explain the former importance of the copyright publications as a reason for their choosing.

2. Furthermore, it's not clear the non-citation aspects of the election.

a. We assume the reviewer means financial aspects of the choice of copyright type. We agree and wish more information about revenue streams were available.

3. Many times -and this is a serious problem in the present model of researching, dissemination and academic careers- the election of one review or another is focused not only or specifically in the impact but in the prestige -former or not- of each publication. Of course, citations are the way to consolidate the academic and scientific name of a review, but many times researchers are looking for a quick curriculum feedback instead of having a medium to long-term perspective.

a. We agree that authors may other factors than citation counts in choosing a journal to publish, but our method will catch all journals indexed at PubMed, regardless of prestige. No edits made.

b. We may have missed usages such as quick curriculum feedback, and this argument could extend to quick patient surveys, Maybe the unreported usage continues, but if so, did the unreported usages more likely to have used the surveys with restrictive terms? This is an interesting and plausible hypothesis as illegal use may certainly not be published. However, if the unreported usage was more likely with restrictive surveys, it would still support our assertion that blended copyrights may retain revenue as the illegal use is likely by noncommercial entities and unfunded. Thank you for this comment.

---

## [Editor Report · Decision Letter 1]

7 May 2026

Benefits and harms of copyright restrictions and conditions on burnout and other psychometric assessment scales

PONE-D-25-65524R1

Dear Dr. Badgett,

We’re pleased to inform you that your manuscript has been judged scientifically suitable for publication and will be formally accepted for publication once it meets all outstanding technical requirements.

Kind regards,

Piotr Stec

Guest Editor

PLOS One

Additional Editor Comments (optional):

Dear Authors,

Thank you very much for your work and for teh extensive comments on reviewers' remarks.

Yout effort is really appreciated.

Best regards

Piotr Stec

---

## [Editor Report · Acceptance letter]

PONE-D-25-65524R1

PLOS One

Dear Dr. Badgett,

I'm pleased to inform you that your manuscript has been deemed suitable for publication in PLOS One. Congratulations! Your manuscript is now being handed over to our production team.

Kind regards,

on behalf of

Dr. Piotr Stec

Guest Editor

PLOS One